# Composite Hydrogel Microspheres Encapsulating Hollow Mesoporous Imprinted Nanoparticles for Selective Capture and Separation of 2′-Deoxyadenosine

**DOI:** 10.3390/molecules27217444

**Published:** 2022-11-02

**Authors:** Lu Liu, Mengdie Zhou, Jianming Pan

**Affiliations:** 1School of Chemistry and Chemical Engineering, Jiangsu University, Zhenjiang 212013, China; 2Key Laboratory of Functional Molecular Solids, Ministry of Education, Anhui Normal University, Wuhu 241002, China

**Keywords:** 2’-deoxyadenosine, hollow mesoporous nanoparticles, molecularly imprinted polymers, hydrogel microspheres, selective adsorption

## Abstract

Hollow mesoporous silica nanoparticles have been widely applied as a carrier material in the molecular imprinting process because of their excellent properties, with high specific surface area and well-defined active centers. However, these kinds of materials face the inevitable problem that they have low mass transfer efficiency and cannot be conveniently recycled. In order to solve this problem, this work has developed a composite hydrogel microsphere (MMHSG) encapsulated with hollow mesoporous imprinted nanoparticles for the selective extraction of 2’-deoxyadenosine (dA). Subsequently, the hollow mesoporous imprinted polymers using dA as template molecule and synthesized 5-(2-carbomethoxyvinyl)-2′-deoxyuridine (AcrU) as functional monomer were encapsulated in hydrogel. MMHSG displayed good performance in specifically recognizing and quickly separating dA, whereas no imprinting effect was observed among 2′-deoxyguanosine (dG), deoxycytidine (dC), or 5′-monophosphate disodium salt (AMP). Moreover, the adsorption of dA by MMHSG followed chemisorption and could reach adsorption equilibrium within 60 min; the saturation adsorption capacity was 20.22 μmol·g^−1^. The introduction of AcrU could improve selectivity through base complementary pairing to greatly increase the imprinting factor to 3.79. Therefore, this was a successful attempt to combine a hydrogel with hollow mesoporous silica nanoparticles and molecularly imprinted material.

## 1. Introduction

With the rapid development of genetic engineering and research on related drugs, 2’-deoxyadenosine (dA), known as an endogenous anti-aggregating substance that inhibits platelet aggregation in vitro and acts synergistically with other antithrombotic drugs, is widely demanded on the market [1]. There are three main approaches to synthesizing dA [2]. It is mainly prepared by degradation of deoxyribonucleic acid, which leads to many by-products after degradation due to the instability of DNA. It can also be obtained by chemical synthesis or biotransformation. All of these methods face the difficulties of low dA content in the products and many by-products, which seriously restrict pharmacological research and application development of high-quality dA and its derivatives. Therefore, it is of great scientific, economic, and social importance to explore new methods for the selective separation and purification of dA, while new methods for its isolation and purification have rarely been explored.

Molecular imprinting (MI) is a polymerization technique allowing monomers to cross-link while associated around a template molecule that fixes the arrangement of the monomers complementary to the template molecule [3,4]. MIPs prepared by conventional polymerization methods usually have a thick imprinted layer and the template molecules are difficult to elute [5,6,7,8,9]. These problems can be solved by surface molecular imprinting technology [10]. Surface imprinting is a new technique for preparing imprinted materials by controlling the positioning of the template on or near the surface of the material to create more-effective recognition sites [11]. Surface imprinted polymers have higher binding capacity and faster mass transfer kinetics than conventional imprinted polymers. Many particles are used as carriers in the surface imprinting process, such as SiO_2_ [12,13], Fe_3_O_4_ magnetic nanoparticles [14], chitosan [15], and Al_2_O_3_ films [16]. Among them, SiO_2_, as a non-swelling inorganic material, exhibits good chemical and mechanical stability and is stable in weakly acidic environments or at high temperatures. However, SiO_2_ particles tend to be solid and spherical, and surface molecular imprinting sites can only be formed on their outer surfaces, with the result that fewer sites are produced [17,18]. Therefore, the preparation of carrier materials with large specific surface area is crucial to improving the surface imprinting effect.

Hollow mesoporous nanoparticles are materials with a large penetrating pore in the shell wall and a cavity inside [19]. Hollow silica (SiO_2_) nanomaterials are of interest because of their excellent biocompatibility, easy surface functionalization, and good thermal stability [20,21,22,23]. By extending the Stöber method [24], hollow mesoporous SiO_2_ particles can be synthesized without toxic chemicals, making them ideal adsorbent materials. However, facing the diversity and complexity of biological sample matrices, the development of new adsorbents with excellent selectivity remains a challenge. Surface molecularly imprinted polymers (SMIPs) based on hollow-structured nanoparticle carriers are considered to be one of the most important molecular recognition materials for drug delivery, sensors, and separations, as they have the advantage of high specific surface area and specific recognition properties [25]. Moreover, the effect of spatial site resistance can be overcome by this surface imprinting strategy, increasing the number of effective binding sites and thus improving recognition efficiency. In addition to hollow spaces and tailored porosity, hollow materials can be further integrated with other components to introduce new functionalities through surface modification or encapsulation [26,27]. As the size of mesoporous silica is often at the nanometer or micron scale, which is not conducive to separation and collection in practice, it is important to prepare adsorbents that facilitate recovery [28]. Thus, hydrogels are introduced to the experimentation. Hydrogels have been used in various fields, including biomedical, water treatment, and product purification due to their hydrophilic polymer chain with a three-dimensional porous network, which can absorb large amounts of water and store different types of compounds [29,30,31]. For hydrogel adsorbents, their role is also multifaceted [32,33]. On one hand, they can be used as support materials for particles or affinity groups to encapsulate molecularly imprinted polymer particles, thereby enabling easy separation and collection of adsorbents [34,35]. On the other hand, due to their ability to absorb large amounts of water and store different types of compounds, encapsulation of molecularly imprinted polymer particles can greatly improve the mass transfer efficiency of molecular imprinting [35,36,37]. Gelatin is a low-cost, non-toxic natural biopolymer [38] with high availability and easy functionalization. The hydrophilic nature of gelatin has led to extensive research on its hydrogel applications. However, gelatin also has some limitations, such as high solubility at room temperature and poor mechanical properties [39]. In order to improve the quality of gelatin hydrogels, it has been suggested to introduce more parts into the polymer chains and to achieve a higher degree of cross-linking [40]. To date, despite the large amount of literature related to gelatin materials, only a few published studies have utilized gelatin methacryloyl (GMA) as a monomer for emulsion droplets to prepare hydrogel microspheres [41].

Herein, in order to obtain a specific, efficient, and easily recoverable adsorbent, this work proposes to develop composite hydrogel microspheres encapsulated with hollow mesoporous imprinted nanoparticles for the selective extraction of 2’-deoxyadenosine (dA). First, hollow mesoporous SiO_2_ particles (MHSs) with -Cl were synthesized in one step through the Stöber process, using polystyrene spheres as hard templates [23] and hexadecyl trimethyl ammonium bromide (CTAB) for soft templates [42]. Subsequently, the MIPs of dA were grafted onto the inner and outer surfaces of the particles by means of the ARGET ATRP technique, using Cl as initiator, AcrU as functional monomer, and dA as template molecule to obtain MMHS. Then, a water-in-oil (W/O) droplet reactor was constructed to prepare MMHS that could encapsulate more imprinting particles by using GMA as the carrier. Finally, the adsorption kinetics, equilibrium, selectivity, and regeneration performance of MMHSG on dA were investigated via static adsorption experiments, and the feasibility of the selective enrichment of dA from complex matrix samples by the adsorbent was verified through combination with actual sample analysis.

## 2. Materials and Methods

### 2.1. Reagents and Materials

Styrene, potassium, persulfate, cetyltrimethylammonium, dimethyl sulfoxide (DMSO), pentanol, polyvinylpyrrolidone (PVP), 3-chloropropyltrimethoxysilane (CPTMS), cetyltrimethylammonium bromide (CTAB), gelatin, methacrylic anhydride (MA), dichloromethane, (3-aminopropyl)triethoxysilane (APTES), anhydrous calcium chloride, sodium hydroxide, sodium citrate, α-bromoisobutyryl bromide (BIBB), triethylamine (TEA), methacrylic acid (MAA), acetonitrile, ethyl orthosilicate (TEOS), copper bromide monohydrate, ethylene glycol dimethacrylate (EGDMA), and ascorbic acid (VC) were purchased from Aladdin (Shanghai, China). We purchased 2’-deoxyadenosine (dA), 2’-deoxyguanosine (dG), 2’-deoxycytidine (dC), 5’-adenosine monophosphate (AMP), disodium hydrogen phosphate, potassium dihydrogen phosphate, anhydrous ethanol, solution, tetrahydrofuran, ammonia, N,N,N,N,N-pentamethyldiethylenetriamine (PMDETA), hydrofluoric acid, and 3-(methacryloxy)propyltrimethoxysilane (MPS) from Sinopharm Chemical Reagent Co. Ltd. (Shanghai, China). Functional monomer 5-(2-methoxyvinyl)-2′-deoxyuridine (AcrU) with complementary thymidine base was synthesized according to the relevant reference [43]. And the 1H NMR spectrum of AcrU is in Appendix A. Deionized water (DI, 18.2 MΩ cm^−1^) was obtained through a Milli-Q water-purification system. The buffer required for the actual sample adsorption was prepared using Wahaha pure water. All the reagents were not purified before use.

### 2.2. Characterization

All the instruments used in this work are listed in the Appendix A.

### 2.3. Synthesis of Polystyrene (PS)

First, 9.6 g styrene was added into a three-necked flask filled with 80 mL deionized water. After passing N_2_ for 30 min, 0.1 g of initiator KPS was added, and then the flask was sealed and mechanically stirred at a constant temperature of 70 °C for 24 h. Then, the products were collected via centrifugation at 12,000 rpm min^−1^ and cleaned with water and ethanol several times. PS was obtained via vacuum drying.

### 2.4. Synthesis of Hollow Mesoporous Silicon Spheres with Large Pores (MHSs)

First, 0.5 g of PS microspheres was added to a three-way flask containing 30 mL ethanol, 25 mL H_2_O, and 0.2 mL NH_3_·H_2_O for ultrasonic dispersion. Subsequently, a mixture of 4.0 mL water and 2.0 mL ethanol containing 0.24 g CTAB was added to the mixture under magnetic agitation. After stirring for 30 min, 0.1 mL TEOS and 0.1 mL CPTES were quickly added. The reaction system was stirred at room temperature for 6.0 h. With the temperature raised to 80 °C, the reaction continued for 12 h. The products collected via centrifugation were washed with pure water several times and dried in a vacuum at 45 °C to obtain silicon spheres. Then, the product was dispersed in an ethanol solution containing 1.0 mL 36 wt% HCl and stirred at 60 °C for 6.0 h to remove the CTAB. Then the centrifugally collected products were stirred in THF solution at room temperature for 12 h to remove the PS template. The hollow microspheres were recovered by centrifugation, cleaned with THF and ethanol several times, and vacuum-dried to obtain MHS.

### 2.5. Synthesis of MIPs (MMHSs) Based on MHSs

The template molecule (dA, 0.0334 g) and the functional monomer (AcrU, 0.0543 g) were added to a 50 mL single-mouth flask containing a mixture of dimethyl sulfoxide and acetonitrile (2 mL/18 mL, *v/v*. The optimal binding ratio of dA and AcrU was studied in Appendix A. In order to form hydrogen bonds with adenine on dA, the template molecule and functional monomer were self-assembled in the dark at room temperature for 2.0 h. EGDMA (0.1 mL) and MHS (0.1 g) were then added to the flask under magnetic agitation at 30 °C. After 0.5 h, CuCl_2_ (10 mg), PMDETA (0.012 mL) and ascorbic acid (20 mg) were successively added to the mixture, and the reaction was performed at 70 °C for 12 h. The product after centrifugation was washed with methanol to remove the unreacted reducing agent, and a methanol/acetic acid eluent with a volume ratio of 9:1 was used to remove the template molecules. Finally, the purified MMHSs were vacuum-dried at 45 °C for 24 h. Similarly, MHS-based non-imprinted polymers (NMHSs) were synthesized in parallel without the addition of dA; other steps were consistent.

### 2.6. Synthesis of MMHS-Encapsulated Hybrid Hydrogel Microspheres (MMHSGs)

Gelatin methacrylate (GMA) was synthesized as the backbone monomer of the hydrogel. We dissolved 5.0 g gelatin in 50 mL of phosphate buffer solution (pH = 7.4), and then 10 mL of MA was slowly added to the above solution and stirred vigorously at 50 °C for 3.0 h. The reaction was then stopped by diluting the reaction solution five times with PBS solution and dialyzed with distilled water through a dialysis bag (cutoff 8–14 kDa) for 7 days to remove impurities, and the product GMA was prepared after freeze-drying.

Ultrasonic dispersions of 200 mg GMA, 15 mg MMHS, 20 mg MBA, and 6.0 mg HHMP were carried out in 5.0 mL distilled water as aqueous phase and deoxidized for 30 min. In addition, 1.0 mL Span 80 was added to a three-neck flask containing 20 mL of olive oil, and the water phase was slowly dropped into it under intense agitation and stirred for 10 min to form a uniform oil-in-water (W/O) emulsion. The flask was then removed and cooled to room temperature in an ice bath. Finally, the emulsion was transferred to the surface dish and placed in UV lamp radiation of 254 nm for 2.0 h at 25 °C, and then centrifuged and MMHSGs collected. The olive oil was cleaned with dilute NaOH solution and washed with water several times. Finally, the product was obtained via freeze-drying.

### 2.7. Batch Mode Absorption Experiments

Binding equilibrium experiments, binding kinetic experiments, and adsorption selectivity experiments were carried out according to the methods mentioned in Appendix A.

### 2.8. Regeneration Tests

The adsorption/desorption experiments were carried out for four cycles to check the regeneration of the MMHSGs. First, 2.0 mg MMHSG were added into 5.0 mL centrifuge tubes, each of which contained 2.0 mL of PBS solution (pH = 7.4, 50 mM) with 300 μmol·L^−1^ of dA. After 2.0 h of shaking, the concentration of dA in the solution was detected and the adsorption capacity *Q*_e_ (μmol·g^−1^) was calculated. After each adsorption, the adsorbed MMHSGs were eluted with acetic acid/water solution (9:1, *v*/*v*) at room temperature for 4.0 h. The regenerated adsorbent was then washed to neutrality with deionized water and used for the next adsorption/desorption cycle after drying.

### 2.9. Actual Sample Analysis

Human urine samples were provided by three male volunteers. The sera were centrifuged at 5000 rpm min^−1^ for 10 min and filtered through a microporous nitrocellulose membrane (pore size 0.22 μm) to remove other impurities. To prepare the actual serum samples, 0.2 mL of serum solution was added to 1.8 mL of PBS solution (pH = 7.4). Then, 5.0 mg adsorbent was added to 2.0 mL of the actual serum sample and the spiked serum sample. After adsorption for 2.0 h, the concentration of each sample in the filtrate was determined by HPLC. The mobile phase consisted of phosphoric acid buffer (pH = 7.4) with a volume ratio of 85:15 and methanol. The flow rate of the mobile phase was kept at 0.8 mL min^−1^, and the column temperature was no more than 30 °C.

## 3. Results and Discussion

### 3.1. Preparation and Recognition Mechanism of MMHSG

The synthesis process of MMHSG is shown in Figure 1. Firstly, hard template PS with negative charge was synthesized using soap-free emulsion polymerization. Then, the negatively charged PS templates were dispersed in the synthesis system, and the cationic surfactant CTAB was combined to form PS@CTAB with positively charged surface properties. At the same time, TEOS and CPTMS were added to the synthesis system using the Stöber method. The silane coupling agent was hydrolyzed in an ethanol/water mixture, and ammonia was used as the catalyst. Negatively charged silica nanoparticles were formed [44]. Through electrostatic interaction, negatively charged silica nanoparticles were assembled on the surface of PS@CTAB and further formed PS@CTAB@Cl@SiO_2_. In other words, the cationic surfactant CTAB acted as a connector between the negatively charged silicon sphere shell and the PS template sphere through electrostatic interaction, enabling SiO_2_ to grow and cover the surface of the PS template sphere. Finally, the synthesized PS@CTAB@Cl@SiO_2_ complex was heated and stirred in acetic acid and THF to remove the PS template balls and form the final product, MHS. Subsequently, MHS with -Cl on the surface was grafted to the MHS surface by ARGET ATRP with the MIP layer formed by AcrU as functional monomer and dA as template molecule to obtain MMHS. Finally, the optimal proportion of hydrogel microspheres (MMHSGs) was obtained via adjusting the size of the droplet reactor by changing the amount of emulsifier and the rotational speed.

### 3.2. Characterization of MMHSs and MMHSGs

To investigate the morphology and chemical properties of sorbents, the SEM, TEM, BET, and XPS analyses were synergistically conducted. As shown in the SEM figure of Figure 2a, it can be found that the particle size of a PS sphere is about 630 nm, with uniform size and high dispersion. The PS@CTAB@Cl@SiO_2_ complex is different from the monodisperse, uniform, and smooth PS template ball. Although it also presents a relatively uniform spherical morphology, its rough surface means that SiO_2_ deposition is coated on the PS template ball and the size increased to 950 ± 50 nm (Figure 2b). The final product, MHS, was obtained by removing the PS template, and its morphology was also characterized by SEM and TEM, as shown in Figure 2c. It can be observed that each nanosphere (MHS) has a large (150–200 nm) opening with a hollow structure (Figure 3a) inside and a neat convex surface. The particle size of these spheres was 680 ± 20 nm, which is smaller than that of PS@CTAB@Cl@SiO_2_, because the particle size of the PS spheres shrank after etching. In order to study whether MIPs were grafted to the surface of MHSs, TEM characterization of MHSs and MMHSs was carried out, as shown in Figure 3. A number of particles and the internal and external surfaces of a single particle were selected at 200 nm, 50 nm, and 20 nm scales for comparison. It can be found that the shell layer of MMHSs (Figure 3d–f) is significantly thicker than that of MHSs, the outer surface of MMHSs have a concave and convex feeling of the polymer layer, and the inner surface of MMHSs also clearly show the point-imprinted polymer. The results prove that MIPs did modify the inner and outer surfaces of the MHSs.

Since MMHSG was synthesized via droplet reactor, the W/O emulsion droplet was observed under a microscope with a uniform size of about 30 μm (Figure 4a). The hydrogel microsphere droplet obtained after 2.0 h of photoinitiated polymerization is shown in Figure 4b. The droplet was obviously condensed into clusters and its size was reduced to less than 20 μm. In order to verify whether MMHSs were encapsulated into MMHSGs, MMHSs labeled with FITC were also dispersed into the aqueous phase and stirred to obtain the emulsion, and then the product was observed with a fluorescence microscope. It can be seen that the labeled MMHSs emit strong green fluorescence (Figure 4d) in the dark field, and do not appear in the droplets (Figure 4c) in the bright field, indicating that MMHSs were successfully encapsulated in the swelling hydrogel microspheres.

The hollow structure of MHS was revealed by BET analysis (Figure 5a). MHSs follow a type IV isotherm, and N_2_ adsorption increases sharply when P/P_0_ approaches 1, indicating the formation of large pores due to hollow structures [45]. In addition, the pore size distribution shows that the sample presents a narrow mesoporous size distribution, with a peak of about 5.0 nm. These mesoporous samples are derived from CTAB, a cationic surfactant, which acts as a pore-forming agent in the synthesis process. BET surface area and the pore volume of MHS were calculated as 169.8 m^2^·g^−1^ and 0.1649 cm^3^·g^−1^, respectively, according to adsorption isotherms.

As shown in Figure 5b, chemical functional groups on MHS, MMHS, and MMHSG surfaces were studied by FTIR spectroscopy. The characteristic peaks of Si-OH, Si-O-Si, and C-Cl at 2950 cm^−1^, 1095 cm^−1^, and 720 cm^−1^ could be observed in the MAP of MHS [46], indicating the successful preparation of hollow silica with Cl on the surface. After polymerization, the characteristic peak of -C-Cl in MMHS disappeared, and two new strong peaks appeared, which were respectively attributed to O=C-NH stretching vibration at 1658 cm^−1^ and C=O stretching vibration at 1490 cm^−1^. In addition, the signals of -OH and -N-H at 3340 cm^−1^ were enhanced and shifted, which confirmed the successful grafting of MIPs to the surface of MHSs. Finally, the peak strength of -N-H, -OH and C=O decreased due to the low content of MMHSs encapsulated in MMHSGs.

XPS tests were performed on MHS, MMHS, and MMHSG to analyze the chemical bonds on their surfaces. Figure 6a shows the full spectrum of XPS, indicating that MHS mainly contains four elements: C, O, Si, and Cl. There are corresponding strong peaks of C 1s, O 1s, Si 2s, Si 2p, and Cl 2p at 284.7 eV, 531.6 eV, 152.9 eV, and 102.8 eV and 200.0 eV, respectively [47]. Meanwhile, due to the relatively low MMHS content in hydrogel microspheres, the peaks of Si 2s and Si 2p did not appear in the MMHSG spectra. The high-resolution C 1s spectrum of MMHS (Figure 6b_2_) can be fitted into six small peaks compared with the C 1s spectrum of MHS, corresponding to C-H (283.95 eV), C=C (284.56 eV), C-C (284.93 eV), C-OH (285.25 eV), C-O-C (286.2 eV), and O=C-NH (288.75 eV). At the same time, the high-resolution spectra of O 1s were fitted with new peaks of C=O (532.08 eV), C-O-C (532.55 eV), and O-C=O (533.0 eV), which proved the successful preparation of MIPs. Finally, compared with the C 1s spectrum of MMHS, an O=C-OH (288.73 eV) signal on the polymer chain of GMA appeared in the spectrum of MMHS. Due to the introduction of GMA, the peaks at C-C, C=C, C-OH, and O=C-NH all have chemical shifts to a certain extent. The high-resolution spectra of MMHS and MMHSG were verified to be consistent with the above conclusion.

Moreover, thermal decomposition behavior analysis (TGA) was conducted to analyze the content of each component in MMHSG (Figure 7). When the temperature increased from 25 °C to 100 °C, the weight loss of water in MMHSG was about 8%. When the temperature increased to 400 °C, there was no significant loss of MHS and MMHS, which proves the high thermal stability of MHS and MMHS. In addition, 32% loss of MHS between 400 °C and 800 °C was mainly due to the decomposition of Si-OH and -Cl on its surface [48], whereas MMHS lost 8% more on this basis, which should be caused by the decomposition of graft MIPs. Meanwhile, the second weight loss of MMHSG was due to the loss of the hydrogel network with the decomposition of MMHS’s surface imprinted polymers, accounting for about 72%.

### 3.3. Analysis of the Adsorption Kinetic Performance of MMHSG and NMHSG on dA

In order to study the adsorption performance of MMHSG and NMHSG, a comparative study of the adsorption kinetics of MMHSG and NMHSG was conducted in this work. The equilibrium adsorption capacity *Q*_t_ of the adsorbent was calculated by Equation (S1) after the detection of the remaining dA in a certain adsorption time range using UV-vis, and the kinetic data were further fitted using quasi-primary and quasi-secondary kinetic equations to investigate the mechanism and rate control steps of the adsorbent action on dA. The results are shown in Figure 8. The adsorption of MMHSG can be divided into two stages: the first stage is rapid adsorption reaching 75% of the equilibrium adsorption amount at 30 min after the start; from 30 min to 60 min, the adsorption rate slows down until it tends to adsorption equilibrium, when it reaches more than 90% of the equilibrium adsorption amount. This also indicates that the excellent performance of the hydrogel matrix allows the adsorbent to maintain a high level of mass transfer efficiency. In addition, the trends of the adsorption rates of the imprinted and non-imprinted materials remained basically the same, and the equilibrium adsorption capacity of MMHSG was approximately more than three times that of NMHSG, indicating the introduction of a large number of effective binding sites in the imprinted materials. The quasi-primary kinetics and quasi-secondary kinetics adsorption-related parameters of the two adsorbents are shown in Table 1. The comparison of *R^2^* and the theoretical adsorption equilibrium capacity revealed that quasi-secondary kinetics (*R^2^* = 0.99) could better describe the kinetic behavior of MMHSG for dA adsorption. Thus, the main rate-controlling step in the adsorption of MMHSG on dA originates from the chemical interaction between the imprinted recognition site on the surface of MMHSG and dA. *h* values of MMHSG were also better than those of NMHSG, demonstrating that MMHSG has a better affinity for dA, which is mainly attributed to the specific recognition effect of the recognition site and the imprinted cavity. Moreover, larger *k*_2_ values imply a shorter time required to reach equilibrium, which corresponds to the corresponding *t*_1/2_ values [49].

### 3.4. Adsorption Isotherm Analysis of dA by MMHSG and NMHSG

An adsorption isotherm study of the behavior between adsorbents is very important for optimizing practical applications and designing schemes for selective separation of nucleoside compounds. Langmuir and Freundlich isotherm models are the most commonly used isotherm models and are also suitable for fitting the adsorption equilibrium process in this work. Figure 9a shows the adsorption isotherms of MMHSG and NMHSG at 25 °C (pH = 7.4) and the corresponding results of Langmuir and Freundlich model fitting. The corresponding fitting data are shown in Table 2. For MMHSG and NMHSG, compared with the Langmuir model (*R^2^* = 0.911, 0.94), the Freundlich model had a better fitting degree to dA (*R^2^* = 0.96, 0.95), indicating that dA was evenly adsorbed on the surface of MMHSG and NMHSG through multilayer adsorption. It is noteworthy that the maximum adsorption capacity *Q*_m_ of MMHSG calculated by the Langmuir equation is 20.22 μmol·g^−1^, which is better than that of NMHSG (5.44 μmol·g^−1^). The high saturated dA extraction capacity of MMHSG can be attributed to the large number of binding sites specifically recognized on the surface of MMHS. In addition, the calculated *1/n* value is between 0 and 1, and the calculated separation factor *R*_L_ is 0.948, indicating that the adsorption of dA by MMHSG is favorable. Scatchard analysis of isothermal binding data showed that MMHSG and NMHSG could be fitted to obtain two straight lines with different slopes, respectively (Figure 9b), indicating the presence of high-affinity binding sites with specific recognition and low-affinity binding sites with non-specific recognition. According to the data calculated in Table 3, compared with NMHSG, the imprinted sites with high affinity for MMHSG and dA accounted for about 83% of all loci. Meanwhile, the *N*_max_ value of MMHSG (21.6 μmol·g^−1^) was 4.8 times that of NMHSG (4.54 μmol·g^−1^), which is consistent with the results of *Q*_m_ theory (Table 2). In addition, the *K*_aH_ (136.99 μmol·L^−1^) and *K*_aL_ (18.52 μmol·L^−1^) values of NMHSG were less different than those of MMHSG, suggesting that the two binding sites in NMHSG may be non-specific hydrogen bonds and hydrophobic interactions.

### 3.5. Adsorption Selectivity and Regeneration of MMHSG

Adsorption selectivity is also a key factor to be studied in the dA adsorption process, and specific adsorption is the basis of adsorbent extraction of the target substance. In order to explore the specific recognition ability of MMHSG and NMHSG for dA, dC, and dG (with similarly sized but different functional groups) and AMP (with a similar structure and different size) were selected as competitors of template molecule dA for adsorption analysis, and the results are shown in Figure 10a. It was found that the adsorption capacity of MMHSG and NMHSG for template molecule dA (8.612 μmol·g^−1^) was significantly higher than for that of the other three competitors. Compared with NMHSG, the adsorption capacity of MMHSG for dA, AMP, dC, and dG increased by 6.3385, 0.35, 0.26, and 0.44 μmol·g^−1^, respectively. This result indicates that MMHSG has specific recognition ability for dA. In addition, the imprinting factor (*IF*) of MMHSG for dA, dC, dG, and AMP was 3.79, 1.44, 1.37, and 1.69, respectively, indicating that the specific adsorption recognition effect of MMHSG was in the order of dA > AMP > dC > dG. Although they have almost the same molecular structure as dA, the adsorption capacity and selectivity of MMHSG for competitors were still much lower than for that of dA, because their shapes and sizes do not match the size of the imprinted cavity. This indicates that molecular imprinting technology plays a very important role in improving the selective recognition ability of adsorbents, and MMHSG has an excellent selective recognition ability for dA compared with other reported MIPs (Table 4). Figure 10b shows the adsorption capacity of MMHSG to dA in four adsorption/desorption cycles. It can be seen that after four cycles, the adsorption capability of MMHSG decreased slightly compared with the initial amount, which may have been caused by the loss of sites on the surface of the material after multiple cycles of contact between the solution and the material, or it may have been caused by incomplete desorption, which finally remained above 88.38% of the initial adsorption amount. These results indicate that MMHSG has good adsorption and regeneration capacity and has a good application prospect.

### 3.6. Analysis of Actual Samples

Finally, to further consider the practical application value of the adsorbents, the selective recognition and rapid separation of dA in human serum samples by MMHSG and NMHSG were investigated, and the dA molecules in solution were detected by HPLC. The results of the tests on the serum stock solution, spiked samples, and samples after adsorption with MMHSG and NMHSG are shown in Figure 11. It can be seen that the dA molecules in the serum stock solution were very small, and the dA molecules in the spiked samples had obvious peaks at about 3.5 min. After adsorption, the peak area of dA molecules was also significantly reduced, and the adsorption capacity followed the order of spiked human serum samples > samples extracted by NMHSG > samples extracted by MMHSG. This result confirms that MMHSG still has good selective recognition and separation ability even in complex real samples. Therefore, MMHSG is a functionalized sorbent with cost-effective value and promising application.

## 4. Conclusions

In summary, SiO_2_ particles (MHSs) with mesoporous and macroporous pore walls and -Cl surface were synthesized in one step by hydrolysis of silane coupling agents TEOS and CPTMS through the Stöber process, using PS spheres as hard templates and cationic surfactant CTAB for soft templates. Subsequently, the MIPs of dA were grafted onto the inner and outer surfaces of the particles via the ARGET ATRP technique, using Cl as initiator, AcrU as functional monomer, and dA as template molecule to obtain hollow mesoporous silica particles (MMHSs). Similarly, a water-in-oil (W/O) droplet reactor with GMA as the carrier was used to explore the water-to-oil ratio and the amount of emulsifier to prepare a hybridized hydrogel microsphere adsorbent (MMHSG) that could encapsulate more imprinted particles, thus increasing the adsorption capacity. The experimental results showed that the adsorption of dA by MMHSG followed chemisorption and could reach the adsorption equilibrium within 60 min; the adsorption equilibrium law for dA at 298 K was in accordance with the Freundlich equation and the saturation adsorption capacity was 20.22 μmol·g^−1^. The introduction of AcrU could improve selectivity through base complementary pairing to greatly increase the imprinting factor to 3.79. The introduction of hydrogels not only ensures mass transfer efficiency, but also facilitates recovery. Furthermore, the combination of hydrogel and molecular imprinting techniques provides a new idea for the synthesis of efficient and specific adsorbents.

## Figures and Tables

**Figure 1 molecules-27-07444-f001:**
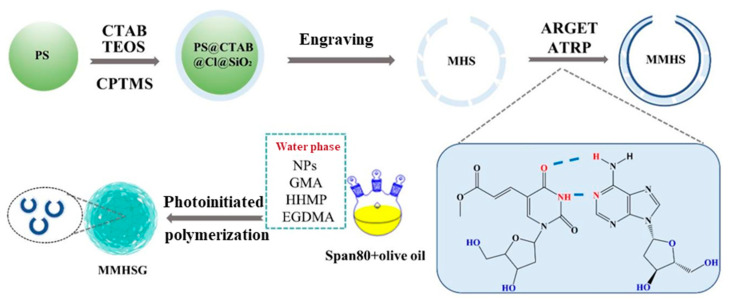
Synthesis mechanism of composite hydrogel adsorbent encapsulating imprinted hollow mesoporous nanoparticles (MMHSGs).

**Figure 2 molecules-27-07444-f002:**
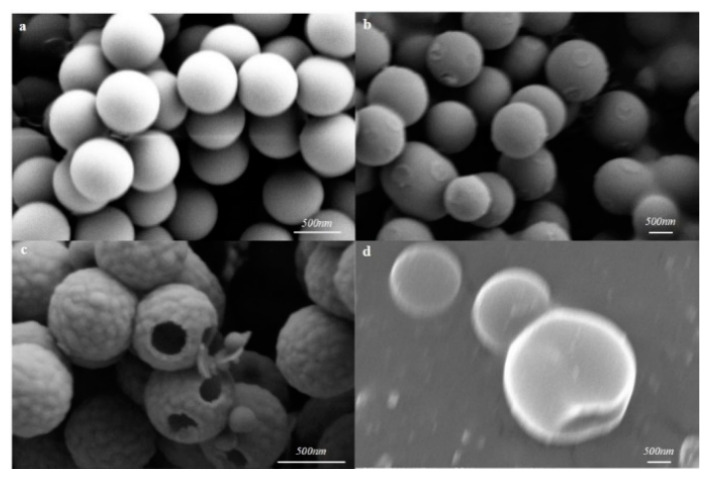
SEM images of PS (**a**), PS@CTAB@Cl@SiO_2_ (**b**), MHS (**c**), and MMHSG (**d**).

**Figure 3 molecules-27-07444-f003:**
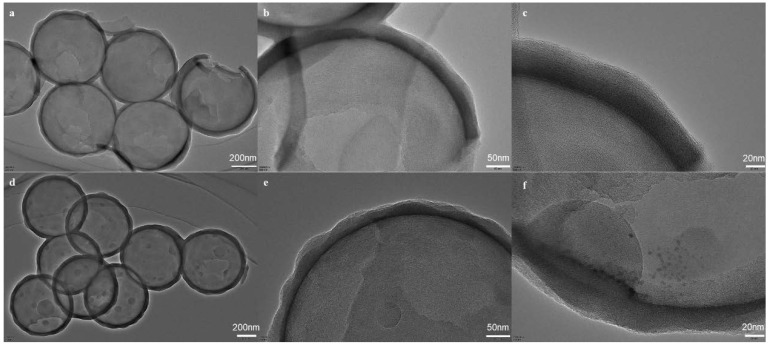
TEM images of MHS (**a**–**c**) and MMHS (**d**–**f**).

**Figure 4 molecules-27-07444-f004:**
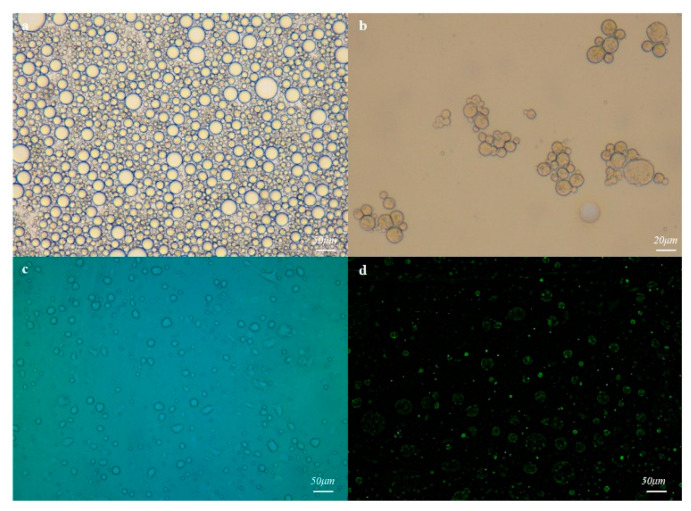
High-resolution microscopy of the W/O emulsion droplet reactor (**a**), micrographs after polymerization (**b**), fluorescence micrographs of FITC-labeled MMHSs under brightfield (**c**), and under darkfield (**d**).

**Figure 5 molecules-27-07444-f005:**
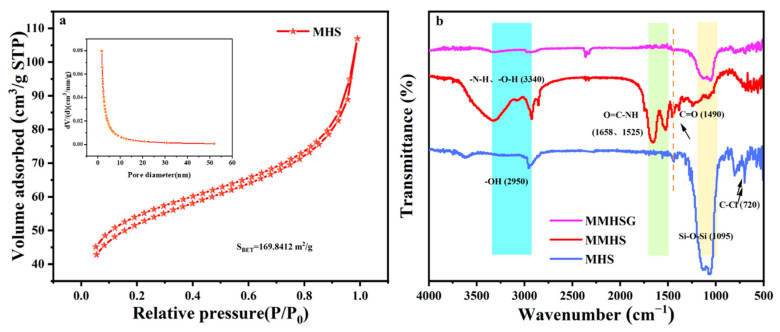
The N_2_ adsorption and desorption isotherms (pore size distribution) of MHS (**a**) and FTIR spectra of MHS, MMHS and MMHSG (**b**).

**Figure 6 molecules-27-07444-f006:**
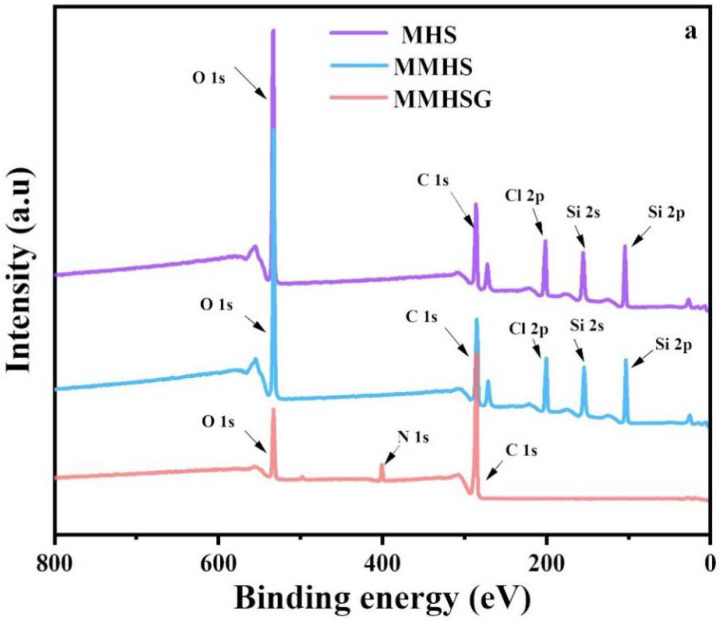
XPS survey spectra of MHS, MMHS, and MMHSG (**a**), and high resolution XPS spectra of C1s and O1s from MHS (**b_1_**,**c_1_**), MMHS (**b_2_**,**c_2_**), and MMHSG (**b_3_**,**c_3_**).

**Figure 7 molecules-27-07444-f007:**
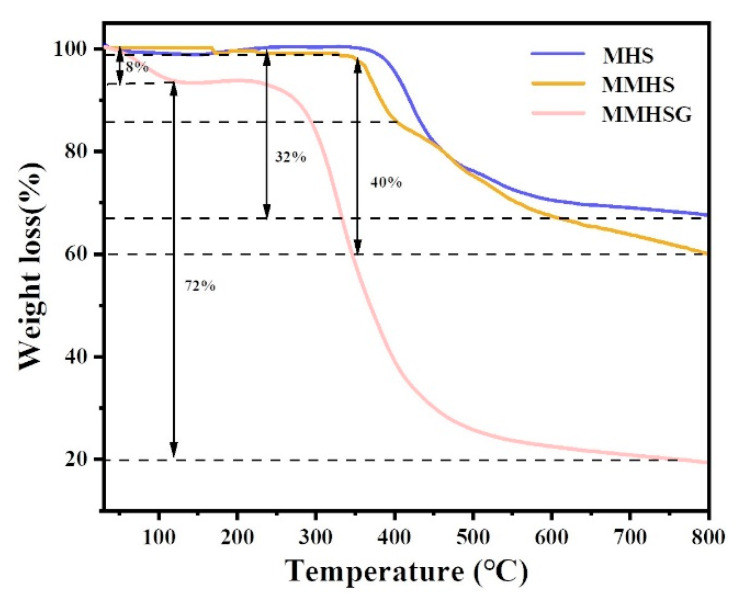
TGA curves of MHS, MMHS, and MMHSG.

**Figure 8 molecules-27-07444-f008:**
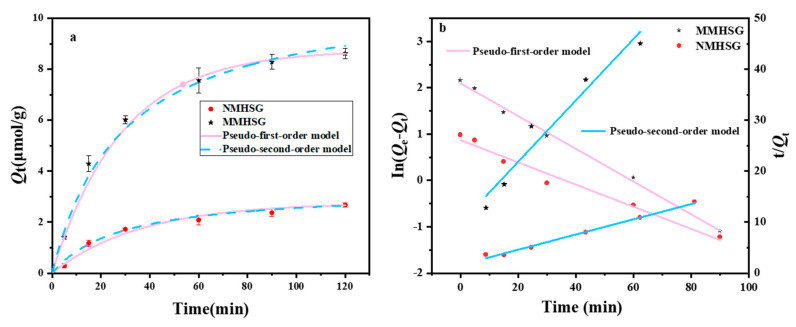
Non-linear kinetic model fitting curves of BHPN@MIPs, BHPN@NIPs, and BHPN for dA (**a**), linear kinetic model evaluation on dA (**b**).

**Figure 9 molecules-27-07444-f009:**
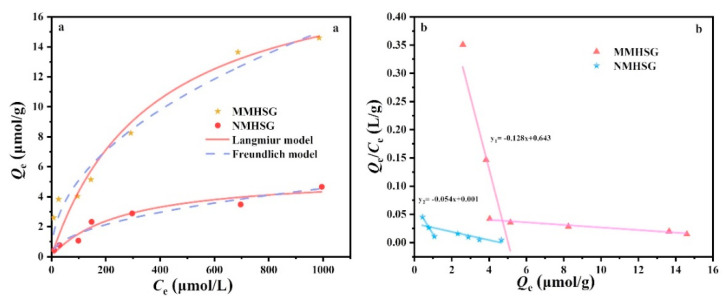
Non-linear Langmuir and Freundlich model fitting curves of MMHSG and NMHSG (**a**), and Scatchard analysis of the binding isotherms (**b**).

**Figure 10 molecules-27-07444-f010:**
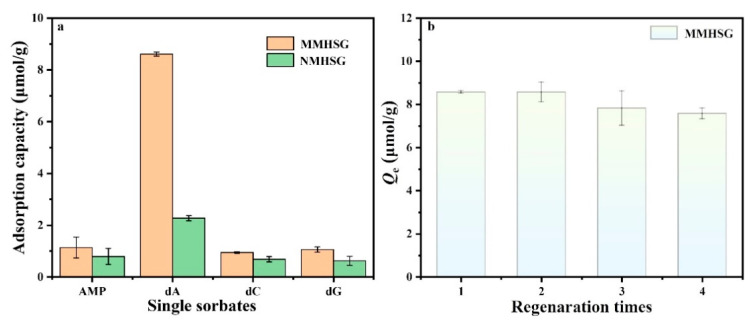
Selectivity adsorption capacity of MMHSG and NMHSG for dA, dG, dC, and AMP (**a**), and regeneration analysis of MMHSG via four sequential adsorption/desorption cycles (**b**).

**Figure 11 molecules-27-07444-f011:**
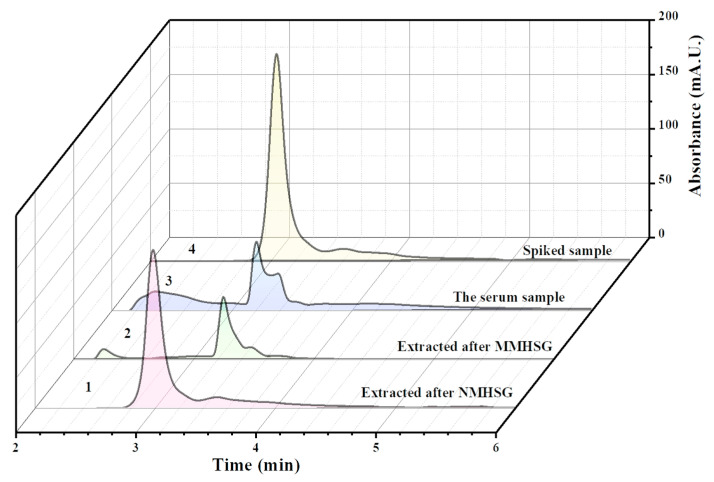
HPLC chromatogram analysis of dA-spiked human serum samples.

**Table 1 molecules-27-07444-t001:** Adsorption kinetic constants and linear regression values from two models onto MMHSG and NMHSG.

Adsorbent		Pseudo-First-Order Kinetic Model			Pseudo-Second-Order Kinetic Model				
	Nonlinear				Nonlinear				
	*Q^a^*_e,e_ (μmol·g^−1^)	*Q^b^*_e,c_ (μmol·g^−1^)	*k*_1_ (min^−1^)	*R* ^2^	*Q^c^*_e,c_ (μmol·g^−1^)	*k*_2_ × 10^−3^ (g·μmol^−1^·min^−1^)	*R* ^2^	*h* (μmol·g^−1^ min^−1^)	*t*_1/2_ (min)
**MMHSG**	**8.615**	**8.48**	**0.042**	**0.99**	**10.39**	**4.19**	**0.99**	**0.45**	**22.86**
**NMHSG**	**2.665**	**2.55**	**0.035**	**0.98**	**3.23**	**10.51**	**0.98**	**0.11**	**29.48**
**Adsorbent**	**Linear**				**Linear**				
	***k*_1_ (min^−1^)**	***Q*_e_ (μmol·g^−1^)**		** *R* ^2^ **	***k*_2_ × 10^−3^ (g·μmol^−1^·min^−1^)**		***Q*_e_ (μmol·g^−1^)**		** *R* ^2^ **
**MMHSG**	**0.035**	**8.141**		**0.99**	**3.58**		**10.648**		**0.99**
**NMHSG**	**0.024**	**2.385**		**0.97**	**10.85**		**2.639**		**0.95**

**Table 2 molecules-27-07444-t002:** Adsorption equilibrium constants for Langmuir and Freundlich isotherm equations.

Adsorbent	Langmuir Isotherm				Freundlich Isotherm		
	*Q*_m_ (μmol·g^−1^)	*K_L_* (L·μg^−1^)	*R* _L_	*R* ^2^	*K*_F_ (μmol·g^−1^) (L·μmol)^−1/n^	*1/n*	*R* ^2^
**MMHSG**	**20.22**	**0.0027**	**0.948**	**0.91**	**0.58**	**0.47**	**0.96**
**NMHSG**	**5.44**	**0.0038**	**0.979**	**0.94**	**0.47**	**0.47**	**0.95**

**Table 3 molecules-27-07444-t003:** Scatchard analysis of the binding isotherms.

Adsorbent	High-Affinity Binding Sites		Low-Affinity Binding Sites	
	*k_a_* (μmol·L^−1^)	*N_max_* (μmol·g^−1^)	*k_a_* (μmol·L^−1^)	*N_max_* (μmol·g^−1^)
**MMHSG**	**442.27**	**21.6**	**7.81**	**4.54**
**NMHSG**	**136.99**	**4.54**	**18.52**	**1.26**

**Table 4 molecules-27-07444-t004:** Comparison of the imprinting factor of MMHSG toward dA with the other reported MIPs.

MIPs	IF	Reference
J-HNPs-MIPs@Gel	1.73	[50]
PC-MIPs	2.71	[51]
J-SNs-MMIPs	1.57	[43]
J-SNs-MMIPs-Pickering	1.499	[52]
MMHSG	3.79	This work

## Data Availability

Not applicable.

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
