# Peer review of "Composite Hydrogel Microspheres Encapsulating Hollow Mesoporous Imprinted Nanoparticles for Selective Capture and Separation of 2′-Deoxyadenosine"

_molecules, 2022, doi:10.3390/molecules27217444_

Round 1

Reviewer 1 Report

The manuscript reports a strategy that develop a composite hydrogel microsphere (MMHSG) encapsulated with hollow mesoporous imprinted nanoparticles for the selective extraction of 2'-deoxyadenosine (dA). This manuscript encompasses valuable scientific work. On the whole, the topic is very interesting, the writing is clear, and the manuscript does not contain technical errors. There are, however, several minor points the authors need to consider:

1.     Check and revise grammatical errors and spelling errors in the manuscript.

2.     5-(2-Carbomethoxyvinyl)-2′-deoxyuridine (AcrU) as the important functional monomer for preparing MIPs should be interpreted in details.

3.     Fig. 8b and Fig. 9b have not been discussed in the whole manuscript.

4.     The performance of MMHSG in actual samples should be mentioned in conclusions.

5.     The formats of all cited references should be corrected according to the formats of Molecules.

Reviewer 2 Report

Paper describes the synthesis of hydrogel microspheres encapsulating hollow mesoporous imprinted nanoparticles, and its application to extract 2′-deoxyadenosine. It is a very interesting strategy to obtain hollow particles of MIP, and the characterizations attest the efficiency of the synthesis’s protocols. Adsorption capacity of imprinted particles were higher than non-imprinted particles, attesting the possible presence of selective binding sites.  A selectivity study with analog molecules demonstrated the high selectivity of the imprinted material for the template. Regeneration studies attested the possibility to reuse the material for, at least, 4 cycles with the same performance. 

However, some important points need to be improved/clarified.

1) A comparison of the proposed MMHSG need to be carried out against a conventional MIP, in terms of regeneration efficiency, selectivity and time for to reach the adsorption equilibrium. The hypothesized advantages proposed by the authors need to be experimentally attested.

2) Selectivity studies were carried out with a samples fortified with the template. However, the study should be also carried out with sample fortified with analog molecules. 

3) Error functions should be used in association with r2 to select the best models for kinetics and isotherm studies.   

Round 2

Reviewer 2 Report

The manuscript has been improved according to the suggestions and can be accepted for publication in this form.